# COVID-19 Vaccination Hesitancy in Mexico City among Healthy Adults and Adults with Chronic Diseases: A Survey of Complacency, Confidence, and Convenience Challenges in the Transition to Endemic Control

**DOI:** 10.3390/vaccines10111944

**Published:** 2022-11-17

**Authors:** Miguel Ángel González-Block, Emilio Gutiérrez-Calderón, Elsa Sarti

**Affiliations:** 1Facultad de Ciencias de la Salud, Universidad Anáhuac, Naucalpan de Juárez 52786, Mexico; 2Facultad de Ciencias UNAM, Ciudad de México 04510, Mexico; 3Sociedad Mexicana de Salud Pública, Ciudad de México 11590, Mexico

**Keywords:** COVID-19, vaccination, vaccine hesitancy, Mexico

## Abstract

Background. Monitoring of SARS-COV-2 vaccine hesitancy is important for epidemic control. We measured vaccine hesitancy among healthy adults and adults with chronic diseases after they had been offered the first dose of the vaccine in Mexico City. Methods. An observational cross-sectional study was undertaken among 185 healthy adults and 175 adults living with chronic diseases. Differences in means of variables for confidence, complacency, and convenience were analyzed. Aggregate indicators were constructed and their association with socioeconomic and demographic conditions and vaccination acceptance analyzed using multivariate analysis of variance and multivariate logistic analysis. Results. Up to 16.8% of healthy adults and 10.3% of sick adults reported not having received the SARS-COV-2 vaccine. Healthy adults were more complacent about COVID-19 risks than adults with chronic diseases, while no differences were found between the two groups regarding other hesitancy aggregate indicators. Among adults with chronic diseases, those with more education and enrolled with a social insurance institution were less complacent of COVID-19, while education was positively associated with convenience across both groups. Less complacency with COVID-19 and more confidence in the vaccine were associated with higher vaccine acceptance across both groups. Among adults living with chronic diseases, the odds ratios of vaccine acceptance were higher for less complacency (OR = 2.4, *p* = 0.007) than for confidence (OR = 2.0, *p* = 0.001). Odds ratios of vaccine acceptance in these two hesitancy indicators were similar among healthy adults (OR = 3.3, *p* = <0.005) and higher than for adults with comorbidities. Conclusions. Confidence in the vaccine and complacency regarding COVID-19 risks play an important role for vaccine acceptance in Mexico City, particularly among healthy adults. The perception of risk regarding COVID-19 is more important than confidence in vaccine safety and effectiveness. Promotion of COVID-19 vaccines needs to focus on decreasing complacency with COVID-19 and increasing vaccine confidence, particularly among healthy adults.

## 1. Introduction

COVID-19 is a respiratory disease caused by the SARS-COV-2 virus and is responsible for high mortality and morbidity rates worldwide [1,2]. The disease can be particularly severe in older adults and adults living with chronic diseases [3,4]. Vaccination is one of the most important measures to prevent severe COVID-19, hospitalizations, and mortality [2,5,6]. The Mexican national SARS-COV-2 vaccination policy prioritized occupational risk groups (health personnel), chronic disease comorbidities, pregnancy, and age (≥60 years). However, the vaccination plan followed up to 31 November 2021 omitted prioritizing according to comorbidities, and vaccinations were offered to the general population based on age-groups only [7,8].

The decision-making process followed by the population to get vaccinated is immersed in a social context of beliefs and perceptions as well as considerations of the availability of the vaccine and its costs [9]. The World Health Organization’s Strategic Advisory Group of Experts (SAGE) proposed the concept of vaccine hesitancy to analyze these beliefs, defined as the delay in the acceptance or the rejection of vaccines despite their availability [10]. Vaccine hesitancy results from a complex interrelation of behavioral and societal factors. Different conceptual models have been proposed to address the complexity, applicability, and potential usefulness of vaccine hesitancy indicators, as well as for the design of surveys and interventions. The “Three Cs” model of vaccine confidence, complacency, and convenience defined by McDonald and collaborators is one of the most useful given that it is intuitive and easy to understand and apply [11].

Hesitancy can be of greater significance for the attainment of SARS-COV-2 vaccination coverage targets in comparison to other vaccines given its pandemic character, its high mortality, and the abundance of COVID-19-specific myths and controversies regarding governmental responses to the pandemic [12,13]. SARS-COV-2 vaccine hesitancy was amply documented prior to vaccine authorization. A systematic review of studies in 35 countries worldwide prior to vaccine rollout found an overall willingness to accept a hypothetical vaccine of 66.0%, ranging from 14.4% in Cameroon to 98.1% in Hong Kong [14]. Several studies in Mexico undertaken prior to vaccine authorization or in the first three months of introduction showed diverse levels of hesitancy, albeit applying different methodologies. Ramonfaur and collaborators reported a 15% refusal rate among adults surveyed at the national level when asked about a hypothetical 90% effective vaccine [15] while Carnalla and collaborators reported that 24.8% of adults in Mexico City would refuse a hypothetical vaccine of unspecified effectiveness [16]. Among breast cancer patients, 3% would refuse the vaccine, 4% would be vaccinated only if it became mandatory, and 26% would wait and see if adverse effects affected others [17]. A similar level of hesitancy was expressed among patients with rheumatic diseases, at 35.5% [18].

There is a risk of increased hesitancy as the pandemic leads to national endemic disease control, as suggested by the most recent influenza pandemic. A worldwide systematic review of influenza vaccination intention and behavior studies identified important barriers to vaccination against influenza in all risk groups. The most frequent reasons for hesitancy were low perceived risk of the disease, lack of trust in the authorities, and low perceived safety of the vaccine [19]. González Block and colleagues found significant influenza vaccine hesitancy across risk groups and countries in Latin America, affected particularly by high levels of complacency among adults living with chronic diseases [20].

The role played by confidence, complacency, and convenience in SARS-COV-2 vaccination coverage needs to be monitored across risk groups to identify the specific factors that can be modified to reduce vaccine hesitancy and to maintain or increase coverage. Quantitative measurement of each of the 3C vaccine hesitancy components can facilitate the identification of context-sensitive factors associated with vaccination uptake.

In this article, we address SARS-COV-2 vaccination hesitancy and acceptance among adults in Mexico City once all the population had been offered at least one dose of the vaccine. We explore three critical questions. What are the determinants and the levels of confidence, complacency, and convenience across healthy individuals as against those living with chronic diseases? How are the sociodemographic characteristics of risk-group members related to each of the three components of hesitancy? How does each of the 3C hesitancy components affect vaccine acceptance?

## 2. Methods

Study design and inclusion criteria. An observational, cross-sectional quantitative study was designed to measure vaccination hesitancy and its association with socioeconomic variables and vaccination acceptance among the population in Mexico City with or without associated comorbidities. A face-to-face survey was applied by trained surveyors to a sample of adults. The survey was applied at the household level in the last two weeks of December 2021, when all adults had been offered the opportunity to receive at least one dose of the SARS-COV-2 vaccine. Adults living with chronic diseases included persons 18 years and older who reported at least one underlying health condition, such as hypertension, gastritis, diabetes, cancer and pulmonary, renal, or cardiovascular disease. Exclusion criteria included individuals with major impairment due to illness.

Operationalization of vaccination hesitancy. The Three Cs model was operationalized following the protocol proposed by Wheelock and collaborators for integrating self-reported knowledge, attitudes, and practices into vaccine confidence, complacency, and convenience indicators [10]. Confidence is defined as the degree of trust in the efficacy and safety of the vaccine, in the system that delivers the vaccines, and in the motivations of those who make the decisions to achieve effective access to the vaccines. Lack of confidence can be influenced by misinformation about vaccination risks, by affiliation to anti-vaccine groups or through legitimate concerns regarding vaccine safety and efficacy and trust in government and the pharmaceutical industry. Complacency is defined as the perception that the risk of diseases preventable by vaccination is low, and when vaccination is not considered a necessary or the chief preventive measure. Complacency is affected by the level of knowledge on diseases and vaccines as well as by prejudices regarding vaccines. Convenience is defined as the influence on vaccine acceptance of vaccine availability, affordability, willingness to pay, ability to understand and accept vaccine-related information, health service quality, and the degree to which vaccination services are delivered at a time and place and in a cultural context that is convenient and comfortable [11].

Survey tool. A vaccine hesitancy survey was constructed based on pertinent and reliable vaccine hesitancy questions originally developed and validated by Larson et al. based on a matrix of hesitancy determinants selected through a systematic review of peer-reviewed literature and on the expertise of the SAGE Working Group on Vaccine Hesitancy [10]. Some of these survey questions had been pilot-tested in various world regions, including the Americas, through a questionnaire in Spanish and were further tested in a five-country influenza study in South America [16]. We adapted the questionnaire to COVID-19 vaccine knowledge and prejudices based on a literature review in Mexico and internationally.

A separate questionnaire was designed for healthy individuals and for those reporting at least one chronic disease, both structured in 63 closed and Likert scale questions (available as Supplementary documents). Questionnaires were developed in Spanish and pilot-tested. The questionnaire was applied to healthy adults and adults living with chronic diseases through a household survey.

Sample size and selection. A sample size of a minimum of 175 persons within each group of healthy and sick adults was calculated considering a statistical error of 0.05, confidence of 95%, and an estimated 13% of hesitancy, this last just below the lowest proportions of COVID-19 vaccine hesitancy reported prior to the approval of vaccines [15]. Households were selected through a stratified sampling strategy. Basic Geostatistical Areas (AGEBs) for Mexico City furnished by the National Institute of Statistics, Geography, and Information (INEGI) were selected to include middle and low socioeconomic groups. Following a two-stage random sampling strategy, AGEBs were first selected with a probability proportional to size, and secondly a fixed number of households was selected with equal probability within each AGEB. In each household in the sample, healthy adults and adults living with a chronic disease were selected based on a preliminary questionnaire and observing compliance with quotas for each of the two groups. The protocol was approved by research ethics committees of the Faculty of Health Sciences of Anahuac University (see detailed ethics committee information in Ethics Approval below).

Construction of aggregate indicators. Aggregate indicators for each of the 3C components were constructed through coding questions to each of the 3C indicators and averaging their component variables. Indicators for complacency subcomponents of prejudices, knowledge and risk perception were also constructed (Table 1). The vaccine confidence indicator was constructed based on 6 questions on vaccine safety and efficacy. The vaccine complacency component was constructed based on 18 questions including dichotomous and Likert scale responses as well as a list of up to 15 influenza symptoms. One point was assigned for each correct symptom mentioned. The vaccine convenience indicator was constructed adding values for 5 dichotomous and Likert-type responses related to vaccine recommendation and access. Values within each of the 3Cs and in the case of complacency within each sub-indicator were standardized on a scale of 0 to 10, with greater value for more confidence and convenience and less complacency. Within the complacency subcomponents, a greater value was allocated to less prejudice, more knowledge, and more risk perception of COVID-19 risks.

Statistical analysis. One-way ANOVA pairwise multiple comparison tests were applied to analyze the significance of differences in means across risk groups for each of the hesitancy indicators and for the aggregate indicators, as well as of vaccination status (at least one dose). For the tests for equality of means, normality tests were undertaken, complemented by bootstrapping methods. Multivariate analysis of variance was applied to assess the significance of the association between the 3C components and sociodemographic variables and risk groups. Multivariate binary logistic multivariate regression analysis was applied to assess the association within each group (healthy and with chronic diseases) between vaccine confidence, convenience, and complacency and vaccination status. The processing and analysis of the information was carried out using the IBM SPSS V.24 package.

## 3. Results

A total of 360 households were surveyed in Mexico City, with 185 healthy adults and 175 adults with at least one chronic disease. The survey response rate was 78% of households that were contacted. A total of 178 males and 182 females were included, with males corresponding to 50.3% among those with chronic diseases and 48.6% among healthy adults (Table 2). Mean age was 50.3 years for adults living with chronic diseases, and 37.8 for healthy adults. Overall, 16.7% of respondents reported they had contracted COVID-19. Education levels were higher among healthy adults than for adults living with chronic diseases: 10.8% of the former reported up to primary education as against 21.7% of the latter. Adults living with chronic diseases reported as the most prevalent diseases gastritis (38.3%), colitis (28.6%), hypertension (32.6%), diabetes (26.3%), colitis (25.9%), and depression or anxiety (17.1%).

### 3.1. Confidence

Adults with chronic diseases reported lower confidence in the COVID-19 vaccine only with regard to the perception that not all vaccine brands available in Mexico City are equally safe (38.5% as against 28.0%, respectively). Only between 25.6% and 27.2% of respondents across the two groups perceived vaccines to be very safe, while between 29.3% and 21.8% perceived vaccines were adequately tested before approval (Table 3). Between 27.4% and 34.3% of respondents perceived vaccines to be equally effective at protecting against COVID-19, and between 39.1% and 42.4% perceived all vaccines to be very effective.

### 3.2. Complacency: Risk Perception

Perceptions of a high risk of contracting COVID-19 were higher among adults with chronic diseases, with 45.7%, as against 20.5% among healthy adults. Overall, 60.7% of adults with chronic diseases perceived a risk of complicated COVID-19 due to their underlying conditions. The perception of COVID-19 as a high severity disease was also higher among adults with chronic diseases, with 69.2%, as against healthy adults, with 53.6% of cases. Knowing someone who had died from COVID-19 was similar across both groups, with between 68.5% and 73.6% of cases.

Respondents recognized 13 distinct COVID-19 symptoms, with no significant differences across risk groups. The most frequently recognized symptoms across both groups were fever (65.7–68.8%), shortness of breath (54.3–57.3%) and headache (46.9–50.8%).

### 3.3. Complacency: Knowledge

Having obtained information on the COVID-19 vaccine is reported was about half of respondents, with no significant differences across risk groups (52.0–59.5%). Having been advised against vaccination was reported with a higher frequency among adults with chronic diseases (37.1%), as against health adults (33.0%) (Table 3). A total of eight vaccine brands were mentioned by all respondents. The brands mentioned were Astra Zeneca (69.2%), Sputnik (64.2%), Pfizer (55.3%), Sinovac (30.3%), Cansino (24.4%), Moderna (14.7%), Janssen (9.7%), and Covaxin (5.0%) (not shown in tables).

The vaccine was perceived more as protecting against infection and contracting COVID-19 than against hospitalization and death resulting from the disease, with significant differences across the two groups. Overall, 88.9% of adults with chronic diseases and 78.3% of healthy adults perceived the vaccine as important to prevent disease, but 38.8% and 32.0% fully agreed that vaccines protect against hospitalization, and 33.9% and 24.4% fully agreed that the vaccine protects against death, respectively. Only 8.0% of persons living with a chronic disease perceived the need to be vaccinated given their specific underlying condition. Furthermore, 29.0% of them perceived that the COVID-19 vaccine could pose a particular risk given their underlying condition.

### 3.4. Complacency: Prejudices

The belief that COVID-19 vaccines can have negative effects in the long term was similar across both groups (47.0–47.1%). Those with chronic diseases preferred to see how others fared with the vaccine before being vaccinated, with 49.4% as against 39.0% of healthy adults. There were no differences across groups regarding perceptions that COVID-19 vaccines only benefit the pharmaceutical industry (50.0–49.7%), perceptions that vaccines have a chip to control people (10.2–11.5%), and perceptions that COVID-19 vaccines are a plot by the government to reduce the population (21.3% to 23.0%).

### 3.5. Convenience

Among respondents reporting access to COVID-19 vaccine information, between 71.8% and 78.0% found it easy to obtain. Between 67.3% and 78.0% of respondents found information to be reliable and between 34.1% and 44.1% found information provided by the government to be very clear. Exposure to recommendations in favor of COVID-19 vaccination were similar across the two groups (54.6–63.4%). The most frequent source of recommendations across both groups were members of the immediate family (81.8–86.1%) and friends or neighbors (48.6–55.4%). Health workers were mentioned as sources of recommendations only in between 39.6% and 41.6% of cases. Healthy adults reported with higher frequency than adults with chronic diseases receiving information from government authorities, work, school, and social networks (not shown in tables).

### 3.6. Aggregate 3C Indicators

The aggregate confidence indicator was similar for adults with risk factors and healthy adults and (6.67–6.60) (Table 4). The aggregate complacency overall indicator was also similar for both groups (5.58–5.54), as were the vaccine prejudices and vaccine knowledge indicators. Significant differences were observed only for risk perception, with adults with chronic diseases scoring higher than healthy adults (5.04 as against 4.51). The aggregate score for convenience was also similar across both groups (5.83–6.02).

The frequency of respondents that had not been vaccinated with a first dose of the vaccine but that were willing to receive it was similar across both groups (2.16–3.43%). However, frequency of individuals who had not received the first dose and that reported a refusal to accept it was higher among healthy adults, with 14.59%, as against 6.86% of adults with chronic diseases.

Confidence in the vaccine was not significantly associated with age, education, sex, or social security enrollment across the two groups (Table 5). Among adults with chronic diseases, those with more education and enrolled with a social insurance institution were less complacent about COVID-19 (*p* = 0.004 and *p* = 0.014, respectively), while education was positively associated with convenience across both groups (*p* = 0.02 and *p* = 0.014, respectively).

Multivariate logistic regression analysis showed a significant and positive association across risk groups (*p* < 0.007 or lower) between confidence and complacency in relation to vaccine acceptance, defined as reporting having received at least one dose or reporting no vaccination but willingness to receive a first dose (Table 6). No significant associations existed for convenience in either group. Confidence had a higher odds ratio for vaccination acceptance among healthy adults (OR: 3.344, *p* = <0.002), as against adults living with chronic diseases (OR: 1.967, *p* = 0.001). Complacency also had a higher odds ratio for healthy adults (OR:3.293, *p* = 0.005), as against adults living with chronic diseases (2.393, *p* = 0.007). Complacency had a higher odds ratio for vaccination acceptance in comparison to confidence for adults living with chronic diseases, while both factors had a similar weight among healthy adults.

NS means nonsignificant *p*-values. More educated individuals and individuals with social security enrollment showed a higher value of hesitancy indicators (less hesitant) as shown.

Based on the multivariate logistic regression analysis undertaken, we can estimate that a one-point increase along the 0 to 10 scale of the confidence indicator would lead to an expected increment of 19.2% in the probability of vaccine acceptance across both groups. A one-point increase in the complacency score (less complacency) would lead to an increment of 20.7% and 19.4% among adults living with chronic diseases and healthy adults, respectively.

## 4. Discussion

As far as we could ascertain, our study is the first to measure COVID-19 vaccination hesitancy once all the adult population had had the opportunity to be vaccinated with at least one dose worldwide. Our study was limited to the study of healthy adults and adults with chronic diseases in Mexico City, concentrating on 7.1% of the country’s population, and cannot be generalized to the adult population of the country. However, results are indicative of the situation in urban areas, where over 79% of the population resides. The demographic profile of respondents corresponds broadly to that of the population of Mexico City. A selection bias could have been introduced, given that respondents were asked to participate in a survey to explore their experience with COVID-19 vaccination. While the risk of bias is small, it is uncertain how it could have affected the results. The survey tool was designed based on extensively validated questionnaires measuring vaccine hesitancy and was piloted but not validated in the context of COVID-19 given the highly specific situation of the pandemic at the time of application. No issues were reported during piloting or in full-scale application regarding the capacity of respondents to understand the survey questions.

Our study points to a small but still important gap in the acceptance of the first dose of the COVID-19 vaccine, which was larger among healthy adults, with 16.8% unvaccinated, despite its availability. The gap is important given that vaccination occurred in the middle of the pandemic and could grow as complacency increases in the context of endemic COVID-19. The gap in COVID-19 vaccination responds more to delay in vaccine acceptance than to refusal, suggesting that it could be lowered with appropriate strategies to reduce hesitancy. Around a third of surveyed adults perceive that the various vaccines made available were not equally safe and around a fourth perceived them not to be very effective. Furthermore, only a third of adults felt confident that all vaccines were adequately tested prior to their introduction. We interpreted brand recognition as positive in so far as it may reduce complacency, yet prejudices regarding their national origins and information with respect to differential efficacy could increase complacency [21,22].

Our findings suggest that all adults in Mexico City underestimate the risk of contracting COVID-19 and of suffering severe COVID-19, a problem particularly with healthy adults. Furthermore, most adults give greater importance to the vaccine’s role in preventing transmission than to its role in the prevention of hospitalization due to COVID-19 and death. These perceptions are contrary to actual risks and the expected effectiveness of COVID-19 vaccines in reducing severity and death rather than transmission [23,24].

While adults living with chronic diseases were less complacent about COVID-19 than healthy adults, only 8% of the former recognized their health condition as a particular reason to be vaccinated, while 60.7% perceived a risk of complicated COVID-19 due to their underlying condition. These findings are of concern, given that the odds ratio of hospitalization among adults with at least one chronic disease compared to healthy adults in Mexico was 1.84 and of death 1.99 in the middle of the pandemic, i.e., close to a twofold increase [25]. While adults with chronic diseases tend to be older than healthy adults, we found that hesitancy was not associated with age in neither group.

Our results suggest the recognition of COVID-19 symptoms by adults in general could be improved in order to improve the penetration of campaigns to alert the population.

The frequency of prejudices with the vaccine was high, with close to half of healthy adult respondents perceiving negative effects from the vaccine in the long term or preferring to delay vaccination until they see how others fare. Distrust in government was high, with only a third of respondents agreeing with the statement that vaccine adequately tested prior to their introduction in Mexico.

Only over half of respondents had access to information on the COVID-19 vaccines, although the majority of those with access to information found it trustworthy, even though a minority found government information to be clear. Furthermore, family members and neighbors were the most frequent source of information on the COVID-19 vaccine, with health personnel playing a secondary role.

### Aggregate 3C Indicators

Both groups of adults showed similar aggregate COVID-19 vaccine confidence, complacency, and convenience scores, though regarding complacency, the score for risk perception was higher among adults living with risk factors. Differences found across the two groups of adults due to knowledge of the COVID-19 vaccine were not relevant, as healthy adults were less advised against vaccination but more of them believed that vaccines are important to prevent the disease. In the case of prejudices, the only significant difference was that a slightly higher proportion of healthy adults preferred to see how others fared with the vaccine before accepting it.

Our findings related to complacency, education and social security enrollment among adults living with chronic diseases suggest the importance of addressing this component of hesitancy, particularly among those of them that are less educated and lacking social insurance institutions. We did not find a significant association between sex and any of the three hesitancy indicators, in line with the results of a study of aggregated data for 193 countries [26].

We also noted healthy adults were more likely to accept the vaccine than adults with chronic diseases if they were less complacent about COVID-19 and if they are more confident of the vaccine. These findings are congruent with the low hesitancy found among breast cancer patients and rheumatic disease patients in Mexico [15,16].

Our observation of hesitancy at a peak of the pandemic and once the vaccine had been offered to all adults in Mexico City [8] is similar to those found by Ramonfaur and collaborators prior to vaccine introduction, who reported a 15% refusal rate of a hypothetical 90% effective COVID-19 vaccine in Mexico prior to vaccine introduction [15], and lower than the hesitancy levels found by Carnalla and collaborators, who reported that 24.8% of adults in Mexico City would refuse a hypothetical vaccine of unspecified effectiveness [16]. This evidence suggests that hesitancy in Mexico City could have been lowered if most of the population had become confident in a highly effective vaccine or if complacency was lowered in a context of high COVID-19 morbidity and mortality. This suggestion is supported in the observation of a decrease in hesitancy as vaccination was rolled out, as documented by a systematic review of 73 studies in the USA, with an increase in acceptance from 56.2% in the first study undertaken in December 2020 at the point of vaccine authorization to 82.0% in the last study undertaken in May 2021 [27].

While COVID-19 vaccine hesitancy seems to have decreased in parallel to the vaccination campaign, it is important to address vaccination acceptance in a low-mortality COVID-19 scenario sustained with periodic vaccination boosters and with the introduction of new vaccines against a broader spectrum of potential SARS-COV-2 variants [28,29]. The decrease in COVID-19 mortality and severe morbidity and the relatively low vaccine effectiveness in preventing transmission will likely decrease confidence in the vaccine and increase complacency. There is therefore the need to implement periodic monitoring of the 3C components of vaccine hesitancy, particularly among more complacent healthy adults. As proposed by Al-Amer, periodic monitoring should be undertaken using reliable and comparable frameworks and tools aligned with the well-established recommendations by the SAGE of the WHO to focus on confidence, complacency, and convenience [29].

## 5. Conclusions and Recommendations

Decreasing hesitancy towards COVID-19 vaccination will be key to end the pandemic and to keep endemic COVID-19 under control, as previously happened with the H1N1 pandemic in 2009. However, there is no information at present regarding vaccine hesitancy for endemic COVID-19 situations. There are diverse opportunities and pathways for strengthening confidence in vaccines and lowering COVID-19 complacency. Strategies are particularly required to increase the perception of COVID-19 risks among healthy adults while focusing also on special needs of adults living with chronic diseases. Awareness of the benefits of currently available vaccines to prevent severe COVID-19, hospitalization, and death should be increased.

## Figures and Tables

**Table 1 vaccines-10-01944-t001:** Construction of confidence, complacency, and convenience indicators.

Indicator	Description	Question and Notes on Scaling
Confidence in the COVID-19 vaccine	Level of perception of the efficacy and safety of the COVID-19 vaccine	Do you believe that all vaccines are equally safe or not all are equally safe? (Yes, No)
		Do you think that all vaccines help prevent COVID-19 equally or do they not all help equally? (Yes, No)
		Vaccines, in general, against COVID-19 are very effective (Likert)
		Vaccines, in general, against COVID-19 are very safe (Likert)
		Vaccines, in general, against COVID-19 were adequately tested before being allowed by the Mexican authorities (Likert)
		The government is not concerned about my health (Likert)
Complacency A. Risk of COVID-19	Level of perception of the risk of contracting COVID-19, of complicating health and its severity	What level of risk of contracting COVID-19 do you consider yourself to have? (Likert)
		Do you consider that COVID-19 could have some risk of complications for you (because of your advanced age/because of your illness)? (Yes, No)
		How serious do you consider COVID-19 to be? (Likert)
		Tell me some of the main symptoms of COVID-19. (1) (Yes, No)
		Have you personally known someone who has died from COVID-19? (Yes, No)
Complacency B. Vaccine Knowledge	Level of knowledge of COVID-19 and its vaccine	Have you obtained information about the COVID-19 vaccine? (2) (Yes, No)
		In your opinion, to whom should the COVID-19 vaccine be applied? (Open)
		Has anyone advised you not to get vaccinated against COVID-19? (Yes, No)
		Could you tell me the brands of the different vaccines that are being applied in Mexico? (1) (Open)
		In your opinion, how important are vaccines, in general, against COVID-19 to prevent the disease? (Likert)
		Vaccines, in general, against COVID-19 protect against being hospitalized in case of contracting the disease (Likert)
		Vaccines, in general, against COVID-19 protect against dying in case of contracting the disease (Likert)
		Do you consider that vaccines, in general, against COVID-19 could have some risk for you because of your illness? (Yes, No)
Complacency C. Prejudice about vaccines	Level of prejudice expressed about the COVID-19 vaccine	Vaccines, in general, against COVID-19 can have long-term negative effects (Likert)
		Before getting vaccinated, I prefer to see how others are doing with vaccinations in general (Likert)
		Vaccines, in general, against COVID-19 will only benefit pharmaceutical companies (Likert)
		Vaccines, in general, against COVID-19 have a chip to control people (Likert)
		Vaccines against COVID-19 are a government plot to reduce the population (Likert)
Convenience	Level of convenience perceived in access to the COVID-19 vaccine	How easy or difficult has it been for you to get reliable (credible) information about the vaccine? (2) (Likert)
		In general, how reliable the information about the vaccine is? (Likert)
		Do you consider that the information provided by the government on the COVID-19 vaccine has been clear? (Very, Somewhat, Not at all)
		Has anyone recommended you get vaccinated against COVID-19? (Yes, No)
		Who recommended that you get vaccinated against COVID-19? (3) (Open)

(1). One point was assigned for each correct symptom or vaccine brand mentioned. (2). When no access to information was reported, the response “Very difficult” is inputted to the question on ease of access to information and the neutral “Moderately reliable” to the question on reliability of information. (3). Two points were assigned to the responses “Health personnel,” “Health campaigns,” and Government authorities”; otherwise, one point was assigned.

**Table 2 vaccines-10-01944-t002:** Sociodemographic characteristics of survey participants by country and risk group.

	Sex		Age				Education Level			
	Male	Female	Minimum	Maximum	Average	Median	Up to Primary	Secondary	Technical	Higher Education
Adults living with chronic diseases N = 175	50.3	49.7	18	86	50.3	51.0	21.7	27.4	38.9	12.0
Healthy adults N = 185	48.6	51.4	18	75	37.8	35.0	10.8	23.8	42.2	23.2

**Table 3 vaccines-10-01944-t003:** Differences in mean percentage values of confidence, complacency, and convenience across risk groups.

	Adults with Chronic Diseases	Healthy Adults	*p*-Value †
Confidence in the COVID-19 vaccine			
All vaccines are not equally safe	38.5	28.0	0.041
Vaccines are very safe	25.6	27.2	
Vaccines were adequately tested before approval	31.8	29.3	
All vaccines are not equally effective at preventing COVID-19	34.3	27.4	
All vaccines are very effective at protecting against COVID-19	42.4	39.1	
The government is not concerned about my health *	41.4	38.7	
Complacency about COVID-19—Risk			
Perceives high to very high risk of contracting COVID-19	45.7	20.5	<0.0001
Perceives risk of complications of COVID-19 given chronic illness	60.7	30.7	<0.0001
Perceives high to very high risk of severe COVID-19	69.2	53.6	0.003
Knows someone who died from COVID-19	73.6	68.5	
Complacency—Knowledge on COVID-19 vaccine			
Obtained information on COVID-19 vaccine	52.0	59.5	
Advised against getting the COVID-19 vaccine	37.1	33.0	
COVID-19 vaccines are important to prevent the disease **	88.9	79.3	0.014
Fully agree that vaccines against COVID-19 protect from hospitalization	38.8	32.0	
Fully disagree that vaccines against COVID-19 protect from hospitalization	13.8	15.4	
Fully agree that vaccines against COVID-19 protect from dying	33.9	24.4	
Person with a chronic disease perceives the need to be vaccinated given his/her underlying condition	8.0	12.4	
COVID-19 vaccines could have some risk because of chronic illness	29.0	18.8	0.025
Complacency about COVID-19—Prejudices			
COVID-19 vaccines can have negative effects in the long term *	47.1	47.0	
Prefer to see how others fare with vaccine before accepting it *	49.4	39.0	0.05
COVID-19 vaccines only benefit pharmaceutical companies *	50.0	49.7	
COVID-19 vaccines have a chip to control people *	10.2	11.5	
COVID-19 vaccines are a government ploy to reduce the population *	23.0	21.3	
Convenience in getting COVID-19 vaccine information			
Getting reliable information about the vaccine has been easy ***	78.0	71.8	
The information about the vaccine is reliable *	78.0	67.3	
The information provided by the government on the COVID-19 vaccine was very clear	44.1	34.1	
Someone recommended me getting vaccinated against COVID-19	63.4	54.6	

† *p*-values are shown only for statistically significant difference across risk groups at *p* < 0.05. Tests are adjusted for all pairwise comparisons within a row of each innermost sub-table using the Bonferroni correction * “Agree somewhat and fully agree” responses were aggregated. ** “Somewhat or very important” responses were aggregated. *** “Easy or very easy” responses were aggregated.

**Table 4 vaccines-10-01944-t004:** Comparison across risk groups between the means of individual scores of confidence, complacency, convenience * and the percentage of vaccinated individuals according to vaccination status.

Indicator	Adults Living with Chronic Diseases	Healthy Adults	*p*-Value	Total
Confidence	6.67	6.60		6.63
Complacency (COVID-19 risk perception)	5.04	4.51	<0.0001	4.77
Complacency (Vaccine prejudices)	6.37	6.70		6.54
Complacency (COVID-19 and vaccine knowledge)	5.34	5.42		5.38
Complacency (Overall)	5.58	5.54		5.56
Convenience	6.02	5.83		5.92
Not vaccinated but wants to be vaccinated (%)	3.43	2.16		2.78
Not vaccinated and does not want to be vaccinated (%)	6.86	14.59	0.018	10.83

* The scale can vary from 0 to 10, with a higher score for more confidence and convenience and for less complacency (less prejudice, more knowledge, and greater risk perception). *p*-values are shown only for statistically significant difference across risk groups at *p* < 0.05.

**Table 5 vaccines-10-01944-t005:** Association between sociodemographic variables in relation to the 3C indices by risk group, using multivariate analysis of variance models.

		Confidence		(Less) Complacency		Convenience	
		*p* Value	Direction of Association	*p* Value	Direction of Association	*p* Value	Direction of Association
Age	Adults living with chronic diseases	0.684		0.055		0.385	
	Healthy adults	0.941		0.684		0.369	
Education	Adults living with chronic diseases	0.144		0.004 *	(+)	0.020 *	(+)
	Healthy adults	0.572		0.073		0.014 *	(+)
Sex	Adults living with chronic diseases	0.192		0.116		0.469	
	Healthy adults	0.243		0.106		0.584	
Social security enrollment	Adults living with chronic diseases	0.372		0.014 *	N-Y	0.797	
	Healthy adults	0.096		0.166		0.434	

* Significant *p*-values. More educated individuals and individuals with social security enrollment showed a higher value of hesitancy indicators (less hesitant) as shown. “+” means a direct association. “N-Y” means that persons with social security “Y” have less complacency than persons without social security.

**Table 6 vaccines-10-01944-t006:** Odds ratio of COVID-19 vaccine acceptance (at least one dose or willingness to be vaccinated in case of delay) and vaccine confidence, complacency, and convenience, by risk group. Binary logistic multivariate regression analyses.

	Confidence			(Less) Complacency			Convenience		
	O.R.	C.I.	*p*-Value	O.R.	C.I.	*p*-Value	O.R.	C.I.	*p*-Value
Adults living with chronic diseases	1.967	1.297–2.983	0.001	2.393	1.274–4.492	0.007	0.890	0.579–1.367	0.593
Healthy adults	3.344	1.913–5.846	0.00002	3.293	1.427–7.599	0.005	1.192	0.723–1.964	0.492

OR: odds ratio; CI: confidence interval (95%).

## Data Availability

Not applicable.

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
