# Peer review of "COVID-19 Vaccination Hesitancy in Mexico City among Healthy Adults and Adults with Chronic Diseases: A Survey of Complacency, Confidence, and Convenience Challenges in the Transition to Endemic Control"

_vaccines, 2022, doi:10.3390/vaccines10111944_

Round 1

Reviewer 1 Report

Dear authors, I have read and reviewed with interest the manuscript submitted to the journal Vaccines entitled: "Covid-19 vaccination hesitancy in Mexico City among healthy 2 adults and adults with chronic diseases. A survey of complacency, confidence, and convenience challenges in the transition 4 to endemic control"

The introduction is well written and focuses the current state of the topic on your research question.

The methodology is well documented, the variables to be analyzed in the study are well defined, the statistical analysis is correct and the selection of the sample of people participating in the survey is well detailed, although they must specify that a selection bias is incurred, however.

Can you detail how many patients refused to participate in the study?

The survey, being subjected to adaptive changes to the situation due to COVID 19, has not been validated in previous studies, this fact must be considered in the limitations of the study.

The tables are well designed, results well expressed.

 Congratulations on the work done.

Author Response

Reviewer 1:

Dear authors, I have read and reviewed with interest the manuscript submitted to the journal Vaccines entitled: "Covid-19 vaccination hesitancy in Mexico City among healthy 2 adults and adults with chronic diseases. A survey of complacency, confidence, and convenience challenges in the transition 4 to endemic control"

The introduction is well written and focuses the current state of the topic on your research question.

The methodology is well documented, the variables to be analyzed in the study are well defined, the statistical analysis is correct and the selection of the sample of people participating in the survey is well detailed, although they must specify that a selection bias is incurred, however.

THANKS VERY MUCH FOR YOUR COMMENT. WE HAVE NOW SPECIFIED IN THE DISCUSSION SECTION HOW SELECTION BIASES COULD HAVE AFFECTED RESPONSES.

Can you detail how many patients refused to participate in the study?

WE HAVE NOW RESPONDED TO THIS IN THE RESULTS SECTION.

The survey, being subjected to adaptive changes to the situation due to COVID 19, has not been validated in previous studies, this fact must be considered in the limitations of the study.

WE HAVE ADDRESSED THIS IN THE DISCUSSION SECTION.

The tables are well designed, results well expressed.

 Congratulations on the work done.

Reviewer 2 Report

1.  These investigators surveyed 360 adults in Mexico City.  Approximately 50% of adults were healthy and 50% had chronic disease.  Surveys asked questions regarding confidence, complacency, and convenience regarding COVID vaccination. These indicators were then used to determine association with demographic factors and vaccine acceptance.  Approximately 17% of healthy adults and 10% of chronically ill adults had not received a vaccine or did not intend to receive the vaccine.  Less complacency and more confidence were associated with higher vaccine acceptance in both groups.
2.  It is unclear to this reviewer as to how this survey was administered.  Did the survey workers ask the questions to the participants?  Did the survey worker leave the questionnaire and ask the participant to fill it out?  Were the educational levels adequate to understand the questions in the survey so that the answers are reasonably accurate?
3.  The authors should provide more information regarding the calculation of aggregate scores.  For example are the range of possible scores for confidence, for complacency, and for convenience?  The foot note to table 5 should be reviewed.  It requires a more thinking than the average reader will have time for.  The foot note to table 4 also needs attention.  Exactly what does scheme 0 mean?  The discussion seems slightly long.  In fact, the vaccination rate in Mexico City is relatively high.  It seems important. Some sentences in the discussion are confusing.  For example "less complacency is negatively associated to education ".  In addition, they introduce the idea of endemic COVID-19 infections and vaccinations at the end of the discussion.  This may be interesting and potentially important, but there is no real information available on that situation at present.

Author Response

Reviewer 2:

  1. These investigators surveyed 360 adults in Mexico City. Approximately 50% of adults were healthy and 50% had chronic disease.  Surveys asked questions regarding confidence, complacency, and convenience regarding COVID vaccination. These indicators were then used to determine association with demographic factors and vaccine acceptance.  Approximately 17% of healthy adults and 10% of chronically ill adults had not received a vaccine or did not intend to receive the vaccine.  Less complacency and more confidence were associated with higher vaccine acceptance in both groups.
  2. It is unclear to this reviewer as to how this survey was administered. Did the survey workers ask the questions to the participants?  Did the survey worker leave the questionnaire and ask the participant to fill it out? 

THANKS VERY MUCH FOR YOUR COMMENT. WE HAVE NOW SPECIFIED HOW THE SURVEY WAS APPLIED.

Were the educational levels adequate to understand the questions in the survey so that the answers are reasonably accurate?

THANKS VERY MUCH FOR YOUR COMMENTS. WE HAVE NOW ADDRESSED THEM IN THE METHODS SECTION.

  1. The authors should provide more information regarding the calculation of aggregate scores. For example are the range of possible scores for confidence, for complacency, and for convenience? 

The foot note to table 5 should be reviewed.  It requires a more thinking than the average reader will have time for.  The foot note to table 4 also needs attention. 

WE HAVE NOW REVISED THE TWO FOOTNOTES TO MAKE THEM MORE EASILY UNDERSTANDABLE.

Exactly what does scheme 0 mean?

WE ARE NOT SURE WHAT YOU ARE REFERRING TO. KINDLY CLARIFY.

The discussion seems slightly long. 

THANKS VERY MUCH FOR YOUR COMMENT. WE HAVE SHORTENED THE DISCUSSION.

In fact, the vaccination rate in Mexico City is relatively high.  It seems important.

WE HAVE NOTED THIS IN THE DISCUSSION.

Some sentences in the discussion are confusing.  For example "less complacency is negatively associated to education ". 

WE HAVE REVISED AND SIMPLIFIED THE DISCUSSION SECTION.

In addition, they introduce the idea of endemic COVID-19 infections and vaccinations at the end of the discussion.  This may be interesting and potentially important, but there is no real information available on that situation at present.

THANKS FOR THE COMMENT. WE HAVE INCLUDED THIS OBSERVATION IN THE CONCLUSIONS.